# MagicPose4D: Crafting Articulated Models with Appearance and Motion Control
## https://magicpose4d.github.io/

## Abstract

With the success of 2D and 3D visual generative models, there is growing interest in generating 4D content. Existing methods primarily rely on text prompts to produce 4D content, but they often fall short of accurately defining complex or rare motions. To address this limitation, we propose **MagicPose4D**, a novel framework for refined control over both appearance and motion in 4D generation. Unlike current 4D generation methods, MagicPose4D accepts monocular videos or mesh sequences as motion prompts, enabling precise and customizable motion control. MagicPose4D comprises two key modules: **(i) Dual-Phase 4D Reconstruction Module** which operates in two phases. The first phase focuses on capturing the model's shape using accurate 2D supervision and less accurate but geometrically informative 3D pseudo-supervision without imposing skeleton constraints. The second phase extracts the 3D motion (skeleton poses) using more accurate pseudo-3D supervision, obtained in the first phase and introduces kinematic chain-based skeleton constraints to ensure physical plausibility. Additionally, we propose a Global-Local Chamfer loss that aligns the overall distribution of predicted mesh vertices with the supervision while maintaining part-level alignment without extra annotations. **(ii) Cross-category Motion Transfer Module** leverages the extracted motion from the 4D reconstruction module and uses a kinematic-chain-based skeleton to achieve cross-category motion transfer. It ensures smooth transitions between frames through dynamic rigidity, facilitating robust generalization without additional training. Through extensive experiments, we demonstrate that MagicPose4D significantly improves the accuracy and consistency of 4D content generation, outperforming existing methods in various benchmarks.

## 1 Introduction

The 4D generation task involves creating a temporal sequence of 3D models of moving objects. Given the difficulty of the general problem, recent approaches have made use of pre-trained models, using prompts to convey the user's intentions. Recently, there has been a significant focus on the use of text/image-prompts to describe appearance and motion Ling et al. (2023); Ren et al. (2023); Zhao et al. (2023). The general pipeline of such existing consists of two stpdf: (i) acquiring static geometry through a 3D generation (e.g., a text/image-to-3D) model Liu et al. (2023), which generates 3D representation such as meshes/implicit fields according to text/image prompts, and (ii) obtaining motion information via a video generation model Blattmann et al. (2023a), which generates video according to text/image prompts. This approach has achieved impressive 4D content results.

However, challenges lie in facilitating users to freely and precisely specify the articulated motion of a non-rigid 3D object and the generation faithfully reflecting the prompts in terms of accurately capturing desired object appearance, geometry, and motion. The main issues with current methods are the following: **(i) Temporal inconsistency in 3D geometry:** Most current video generation models fail to ensure temporal consistency of 3D geometry. This involves additional complexity when objects are articulated because the 3D shape and movement must change mutually consistently during motion. For instance, while using the existing methods for 4D animal generation, we have observed unnatural and implausible configurations, such as the number of limbs of the object changing across different frames. **(ii) Limited to common motions:** Current video generation models perform well

in generating simple, subtle, and common actions, such as walking or small motions like shaking, but they struggle to satisfactorily generate more complex motions, e.g., involving large or uncommon movements such as *"King Kong dancing Hip-Hop"* and *"pig running like a rabbit"* as shown in Fig.5. **(iii) Text is inadequate for accurately describing the details of motion:** As shown in Tab.4, most existing methods use text as prompts to describe the desired motion, with some efforts Bahmani et al. (2024a) allowing trajectory (root body transformation) control. However, they are unable to specify detailed motions, e.g., by showing a real-world animal's movement (say, via a monocular short video).

To address these issues, we propose **MagicPose4D**, which enables detailed control over both appearance and motion. Unlike existing methods, which rely mostly on text descriptions as motion prompts and for which it is difficult to convey complex or rare motions, our approach accepts monocular videos or dynamic mesh sequences as motion prompts. This allows for more precise supervision and faithful 4D generation. MagicPose4D introduces two key modules: **Dual-Phase 4D Reconstruction Module** and **Cross-category Motion Transfer Module**. The first module estimates a sequence of 3D models of the object while also simultaneously estimating motion parameters from the motion prompts, and the second module transfers this motion to the target object, which is generated by a 3D generation model controlled by appearance prompts.

**Dual-Phase 4D Reconstruction Module**: Given the complexity, especially with articulated objects, we divide this module into two phases. First, we use image descriptors (e.g., segments, flow) as 2D supervision. Additionally, we employ relatively less accurate but geometrically informative pseudo-3D models as geometric priors (using image-to-3D models) to provide 3D supervision for 3D reconstruction. This phase does not impose constraints emanating from the articulated nature of the object, e.g., captured in the underlying skeletal structure and plausible changes in it during motion, allowing each part to learn arbitrary rotations ($\mathbf{R}$) and translations ($\mathbf{t}$) while focusing on learning the model's shape. In the second phase, we ensure the physical plausibility of the motion by enforcing kinematic chain-based skeletal movement constraints. Additionally, we propose a Global-Local Chamfer loss, which ensures that the overall distribution of predicted mesh vertices aligns with the supervision while maintaining part-level alignment, without requiring additional annotations. **Cross-category Motion Transfer Module**: This module achieves cross-species motion transfer by mapping the skeleton of one species to another by establishing joint and limb correspondences through a kinematic-chain-based representation of the skeleton. Our motion transfer module is non-training-based. It helps improve generalization and prevents the poor performance seen in many existing methods when tested on data with significant gaps from the training set. Also, we leverage dynamic rigidity Zhang et al. (2024c) to guarantee smoothness between frames, unlike the existing approaches Song et al. (2021; 2023), which perform frame-independent pose transfer.

The following are the main contributions of this paper: **(i) A Novel 4D Generation Framework:** Our new framework leverages monocular videos as motion prompts, providing more accurate and more precisely specifiable 4D action generation. **(ii) Skeleton Based 4D Representation:** By using skeleton-mediated geometric and 3D prior, we achieve more accurate motion estimation and 3D reconstruction, improving the physical plausibility of the generation. **(iii) Global-local Chamfer Loss**: We introduce a novel loss function to better align estimated mesh vertices with the supervisory 3D model overall while maintaining part-level alignment, without additional annotations. **(iv) Cross-category Motion Transfer**: Our cross-category mapping in terms of skeleton based dynamic rigidity representation enables smooth transitions between frames and robust generalization without additional training. **(v) Outperforms SOTA:** Through extensive experiments, we demonstrate that MagicPose4D provides highly accurate 4D content and significantly outperforms existing methods for 4D reconstruction and pose transfer across all three benchmarks that we experiment with.

## 2  RELATED WORK & MOTIVATION

**4D/3D Reconstruction.** Significant advancements in 4D/3D reconstruction include the development of specialized parametric models such as SCAPE Allen et al. (2003) and SMPL Loper et al. (2023a) for human bodies, MANO Romero et al. (2022) for hand movements, FLAME Li et al. (2017) and EMOCA Daněček et al. (2022) for facial expressions, and SMAL Zuffi et al. (2017) for quadruped animals. However, these methods require predefined parameters that include skeletons and skinning weights, limiting their generalization to uncommon species and out-of-distribution data. Recent

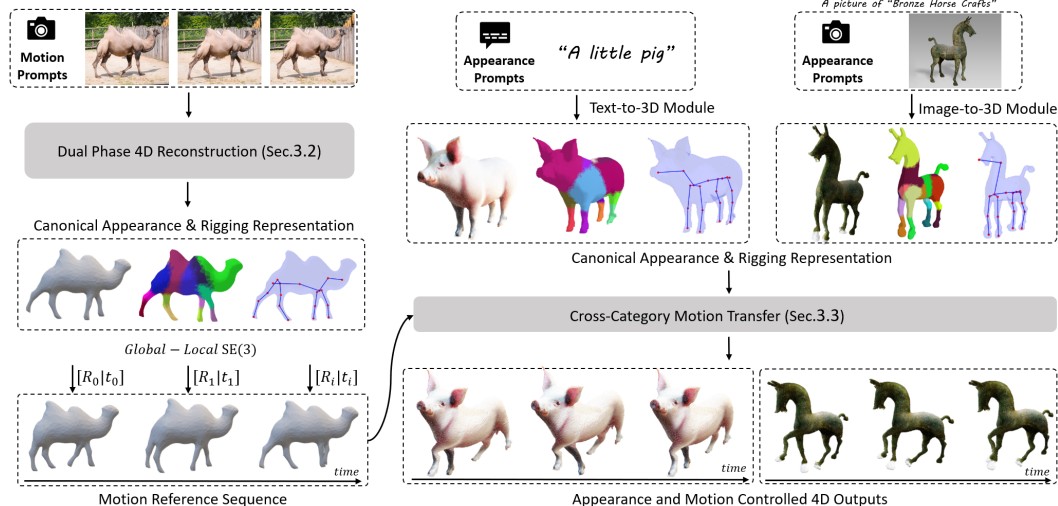

Figure 1: Overview of **MagicPose4D**, which takes motion prompts (monocular video or dynamic mesh sequence) and appearance prompts (text or image) to control the 4D content generation.

neural implicit-based methods Palafox et al. (2021); Yang et al. (2022b); Giebenhain et al. (2023); Zhang et al. (2024c); Yang et al. (2021b) offer promising alternatives by jointly learning static 3D meshes and time-varying parameters without predefined templates. However, they often fail with monocular videos containing sparse views, limiting their practical applicability. To address this, MagicPose4D introduces an innovative Dual-Phase 4D Reconstruction module, which reduces the requirements from multi-view videos to single-view videos and achieves robust reconstruction.

**Diffusion-based 4D Generation and Pose Transfer.** Recent diffusion-based 4D generation methods have shown promising results by leveraging text-to-3D models followed by text-to-video supervision. These techniques Singer et al. (2023); Ren et al. (2023); Ling et al. (2023); Bahmani et al. (2023); Zheng et al. (2023); Gao et al. (2024); Yang et al. (2024); Jiang et al. (2023); Zhao et al. (2023) have improved the geometry and appearance of generated models but are generally limited to minor movements within a fixed location and rely on text prompts, making precise motion control difficult. Our goal is to create dynamic 4D animations that closely transfer the motion of any given reference real-world object. For pose transfer, recent skeleton-based frameworks Song et al. (2021); Liao et al. (2022) have explored the use of rigging points and key points. However, learning-based methods often struggle with in-the-wild data and significant domain gaps between identities. MagicPose4D introduces a cross-category motion transfer module that supports cross-species transfer while ensuring generalization and maintaining temporal smoothness.

## 3 MAGICPOSE4D

MagicPose4D accepts two distinct types of input prompts: (i) appearance prompts and (ii) motion prompts. Consistent with recent methods Blattmann et al. (2023c); Tang et al. (2023), both images and textual descriptions can function as appearance prompts, delineating the desired object and its visual characteristics. In a departure from existing approaches, MagicPose4D enables users to specify precise motions and trajectories by providing a video/mesh sequence representing the anticipated movement.

As illustrated in Fig 1, MagicPose4D comprises three critical components: (i) the 4D Reconstruction module (Sec.3.2), (ii) the Cross-Category Motion Transfer module (Sec.3.3), and (iii) the Image-to-3D Generation module. Each module is tailored to facilitate distinct aspects of dynamic modeling, enabling adaptive 4D reconstructions that align with user-defined specifications.

### 3.1 TERMINOLOGY AND OVERVIEW

To represent an animated 3D model, our method learns static representations, such as the visible canonical shape $\mathbf{S} \in \mathbb{R}^{N \times 3}$, the underlying skeleton $\mathbf{S_k} = \{\mathbf{J} \in \mathbb{R}^{J \times 3}, \mathbf{B_s} \in \mathbb{R}^B, \mathbf{P_i} \in \mathbb{R}^B\}$, and

the skinning weights $\mathbf{W} \in \mathbb{R}^{N \times B}$. Additionally, it captures time-varying parameters, including the global-local (root body-bone) transformations $\tau = \{\tau_0^t, \tau_1^t, \ldots, \tau_B^t\}$ and the camera parameters $\mathbf{P_c}$. Here, $B$, $N$, and $J$ represent the number of bones, vertices on the mesh surface, and joints, respectively. The transformations $\tau_i^t \in SE(3)$ include $\tau_0^t$ for the root body, with the remaining $\tau_i^t$ for the bones. The skeleton topology is described by $\mathbf{J}$ (joint coordinates), $\mathbf{B_s}$ (bone scale), and $\mathbf{P_i}$ (parent indices of joints).

As illustrated in Fig 1, utilizing a short monocular video as a motion prompt, we introduce a dual-phase 4D reconstruction module (Sec.3.2) designed to predict a sequence of 4D meshes as motion references. Appearance prompts may consist of text descriptions, images, or a monocular video. Depending on the type of input, a corresponding module—text-to-3D, image-to-3D, or dual-phase 4D reconstruction is employed to generate the static 3D representation of the desired object. Subsequently, this static representation along with the motion reference are fed into the cross-category motion transfer module (Sec.3.3). This module adeptly transfers motions from the reference to the target object while ensuring temporal smoothness and consistency.

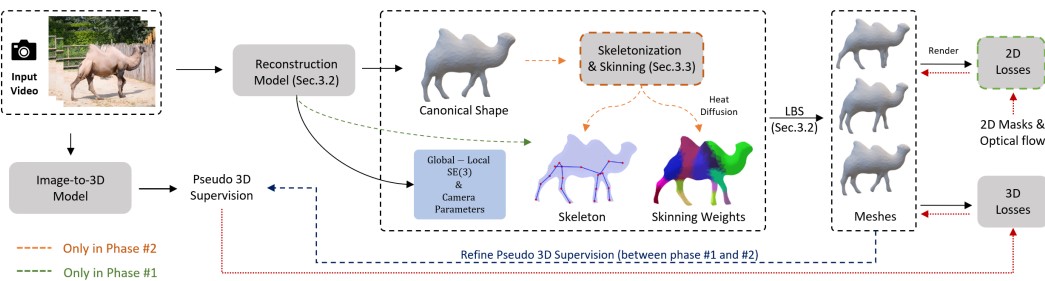

Figure 2: Overview of **Dual-Phase 4D Reconstruction Module**.

### 3.2 DUAL-PHASE 4D RECONSTRUCTION FROM VIDEO

Extracting the 3D motion of an articulated object directly from a monocular short video is extremely challenging. MagicPose4D addresses this issue by reconstructing the object and extracting its motion by learning its physically plausible skeleton.

Reconstructing 4D models from short monocular videos is also a challenging task as it requires jointly learning numerous parameters, resulting in an extensive optimization space. Erroneous predictions of any parameter can trigger cascading failures. To address these challenges, we propose a Dual-Phase 4D Reconstruction module, which employs differentiated supervision across two distinct phases and focuses on learning diverse representations.

In the first phase, the primary focus is on accurately capturing the external appearance (shape) of the model. The underlying skeleton serves merely as an intermediary variable, utilizing non-kinematic chains skeleton and learnable skinning weights to afford the skeleton greater deformation freedom. This approach accelerates the learning of correct shapes. In contrast, the second phase aims to derive a more physically plausible motion reference for effective motion transfer. Therefore, we adopt kinematic chain skeletons and heat diffusion-based skinning weights, which narrow the deformation space of the skeleton, thereby ensuring the plausibility of the internal structure. (Sec.3.2.1)

From a supervision perspective, the first phase blends 2D and pseudo-3D supervision, updating the pseudo-3D supervision at the end of this phase. In the second phase, the 2D loss is removed, relying solely on the updated 3D pseudo-supervision to guide the learning process. (Sec.3.2.2)

#### 3.2.1 MODEL ARTICULATION

**Skinning Weights** $\mathbf{W}$ is designed to represent the probabilities that each vertex corresponds to $B$ semi-rigid parts. **In the first phase**, following Yang et al. (2021b; 2022b), the skinning weights are modeled by the mixture of $B$ Gaussian ellipsoids as: $\mathbf{W}_{n,b} = \mathcal{F}e^{-\frac{1}{2}(\mathbf{X}_n - \mathbf{C}_b)^T \mathbf{Q}_b (\mathbf{X}_n - \mathbf{C}_b)}$, where $\mathcal{F}$ is the factor of normalization and precision matrices $\mathbf{Q}_b = \mathbf{V}_b^T \mathbf{\Lambda}_b \mathbf{V}_b$. In each Gaussian ellipsoid, $\mathbf{C} \in \mathbb{R}^{B \times 3}$ denotes Gaussian centers, $\mathbf{V} \in \mathbb{R}^{B \times 3 \times 3}$ defines the orientation and $\mathbf{\Lambda} \in \mathbb{R}^{B \times 3 \times 3}$

denotes the diagonal scale matrix. $\mathbf{X}_n$ is the 3D location of vertex n. **In the second phase**, as shown in Fig.3, a skeletonization module, is leveraged to obtain a skeleton for the canonical mesh, and following Tong et al. (2012); Baran & Popović (2007), skinning weights $\mathbf{W}$ are obtained by a heat diffusion process. This approach guarantees a more natural assignment of skinning weights to the bones, enhancing the realism of the skeletal animations.

**Blend Skinning.** The mapping from surface vertex $\mathbf{X}_n^0$ in canonical space (time 0 by default) to $\mathbf{X}_n^t$ at time t in camera space is designed by blend skinning. The forward blend skinning is shown below:

$$\mathbf{X}_n^t = \tau_0^t(\sum_{b=1}^{B} \mathbf{W}_{n,b}\tau_b^t)\mathbf{X}_n^0, \tag{1}$$

including the number of bones $B$, skinning weights $\mathbf{W}$ and the transformation $\tau^t = \{\tau_b^t\}_{b=0}^B$ at time $t$, where $\tau_0^t$ represents the root body transformation and $\tau_{b>0}^t$ describes the bone transformation. Each $\mathbf{X}_n^0$ is first transformed by the weighted sum of each bone transformation $\mathbf{T}_{b>0}^t$ and then transformed by root body transformation $\mathbf{T}_0^t$ to achieve $\mathbf{X}_n^t$.

**Skeleton Articulation.** As described above, bone and root body transformations $\tau$ are crucial in the process of blending skinning. **In the first phase**, we define these transformations as independently learnable $SE(3)$ transformations. Practically, this is implemented by initializing learnable quaternions to compute the rotation matrices, along with defining learnable translations to achieve affine transformations. During this phase, we impose no constraints on the skeleton; each bone can rotate and translate freely. These bones serve merely as intermediate variables, with the ultimate objective of accurately determining the correct shape. **In the second phase**, we define per-frame joint angles $Q$, each describing the pose between a bone and its parent with three degrees of freedom. Instead of directly learning bone transformations, we determine $Q$ and compute the bone transformations $\tau_{b>0}^t$ using forward kinematics. The deformed mesh is then obtained by blend skinning.

**Extendable Bones.** It is noteworthy that kinematic-chain-skeleton-based methods rest on a fundamental assumption that the structure of the utilized skeleton closely mirrors the natural skeletal architecture of animals. This allows for the desired deformed mesh to be achieved by manipulating the skeleton's pose and employing blend skinning techniques. However, in practice, this assumption often does not hold completely. Animal bones are not solely rigid hinge connections; they include extendable cartilaginous tissues that lead to non-rigid deformations among joints. To address this, we introduce the concept of extendable bones. We allow for slight variations in bone length between frames, achieved by learning a time-varying scale parameter for each bone $\mathbf{B_s} \in \mathbb{R}^B$. This modification enhances the flexibility and realism of our skeletal model.

### 3.2.2 SUPERVISION AND LOSSES

As shown in Fig. 2, given the input from a monocular video, we utilize segmentation models Kirillov et al. (2023); Zhang et al. (2024a) and optical flow prediction models Teed & Deng (2020) to obtain 2D supervision. Furthermore, we employ an image-to-3D model to independently predict meshes for each frame, serving as pseudo-3D supervision. A question arises: *"Why not directly use an image-to-3D model to independently predict 3D meshes for each frame?"* As illustrated in Fig.6 (c), objects always exhibit self-occlusion in the video, under which circumstances image-to-3D models typically fail to produce accurate results. For instance, some frames might only depict a camel with two visible legs, resulting in a 3D mesh sequence that does not effectively capture the action information portrayed in the video. Moreover, as demonstrated in Video 3 of the supplementary materials, meshes generated directly from the image-to-3D module are independent of one another, thus lacking temporal continuity and smoothness. Although the initial pseudo-3D supervision may not be very accurate, it still provides valuable geometric priors that help address the information loss caused by insufficient perspectives of the object in the video.

In the first phase, we blend both 2D and 3D supervision to optimize the mesh shape and leverage a reconstruction loss, which consists of silhouette loss, optical flow loss, texture loss, perceptual loss, smooth, motion, and symmetric regularizations, and Global-Local Chamfer (GLC) Loss. In the second phase, we only leverage GLC Loss and regularization terms without 2D losses. The details of the 2D loss functions and regularizations will be described in the Appendix.6.6.

**Global-Local Chamfer Loss (GLC).** The objective of GLC loss is to ensure that the predicted mesh closely resembles the expected mesh in **(i) overall shape** and in terms of their **(ii) respective poses**.

This dual focus helps achieve high fidelity in both the structural and positional accuracy of the meshes. The GLC loss is the sum of chamfer distances across two levels. Initially, it involves the computation of the chamfer distance for the entire predicted mesh $\mathbf{S}$ and Pseudo mesh $\hat{\mathbf{S}}$ as follows:

$$\mathcal{L}_{\text{global}}(\mathbf{S}, \hat{\mathbf{S}}) = \frac{1}{|\mathbf{S}|} \sum_{x \in \mathbf{S}} \min_{y \in \hat{\mathbf{S}}} \|x - y\|^2 + \frac{1}{|\hat{\mathbf{S}}|} \sum_{y \in \hat{\mathbf{S}}} \min_{x \in \mathbf{S}} \|x - y\|^2. \tag{2}$$

The second level involves computing the weighted chamfer distances between $B$ parts. First, we calculate skinning weights $\mathbf{W}, \hat{\mathbf{W}} \in \mathbb{R}^{N \times B}$ for $\mathbf{S}$ and $\hat{\mathbf{S}}$ via the heat diffusion process Tong et al. (2012); Baran & Popović (2007). Then, we perform an *argmax* across the $B$ dimension to achieve a part-wise decomposition of the whole body into $K$ parts, where $K$ is less than or equal to $B$. In practice, $K$ often equals $B$, but in cases where $K$ is less than $B$, the loss computation simply omits the non-existent parts. Subsequently, we calculate the part-level Chamfer distances between the $K$ pairs from predicted mesh $\mathbf{P}_k, k \in \{0, ..., K - 1\}$ and Pseudo mesh $\hat{\mathbf{P}}_k$, as shown in the following equation:

$$\mathcal{L}_{\text{local}}(\mathbf{S}, \hat{\mathbf{S}}) = \frac{1}{|K|} \sum_{k=0}^{K-1} \frac{1}{|\mathbf{P}_k|} \sum_{x \in \mathbf{P}_k} \mathcal{W} \min_{y \in \hat{\mathbf{P}}_k} \|x - y\|^2 + \frac{1}{|\hat{\mathbf{P}}_k|} \sum_{y \in \hat{\mathbf{P}}_\mathbf{k}} \mathcal{W} \min_{x \in \mathbf{P}_k} \|x - y\|^2, \tag{3}$$

where $\mathcal{W} = \mathbf{W}_{x,k} \times \hat{\mathbf{W}}_{y,k}$ represents the multiplication of skinning weights between vertices $x$ and $y$ for bone $k$. $\mathcal{W}$ serves as a rough key points estimator based on skinning confidence. It assigns greater weight to vertices at the central part of a segment and lesser weight to vertices at the junctions of multiple parts. This approach effectively addresses inconsistencies at the edges of part decompositions when calculating skinning weights using heat diffusion for models in various poses.

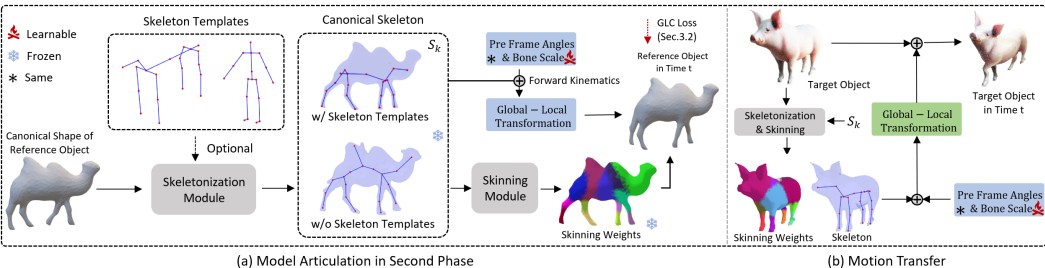

Figure 3: Overview of **Cross-Category Motion Transfer Module**.

### 3.3 CROSS-CATEGORY MOTION TRANSFER

In the second phase of 4D Reconstruction, as shown in Fig. 3(a), we extract the underlying skeletal motion of the reference meshes. Given a sequence of meshes for the reference object, the skeletonization module extracts the skeleton of the canonical shape, which is defined as the shape corresponding to time $= 0$. We fix the canonical skeleton and the skinning weights, thus controlling the skeleton's pose solely by learning the pre-frame angles and bone scales. Subsequently, by employing blend skinning, we obtain the deformed mesh.

The input to the skeletonization module is a mesh, along with an optional skeleton template. When the reference object is a commonly recognized form, such as a quadruped or a human, we utilize the corresponding skeleton template and embed it into the mesh. Following the methodology described in Baran & Popović (2007), we construct a coarse discrete representation to locate the approximate position of the skeleton within the internal space of the mesh. This process involves embedding the skeleton into a graph derived from the character's internal volume and determining an optimal solution using the A* algorithm across all possible matches. When a template of the object is not available, we employ skeleton extraction methods Bærentzen & Rotenberg (2021); Zhang et al. (2024c) to extract the canonical skeleton of the reference object. After learning the reference motion represented by the pre-frame angles and the bone scale of the skeleton, we can readily compute the global-local transformation of the target object using forward kinematics, as illustrated in Fig. 3(b).

We extract the canonical skeleton for the target object by inputting the canonical skeleton of the reference object, adhering to the method previously described following Baran & Popović (2007). Subsequently, blend skinning is employed to generate the deformed meshes.

## 4 EXPERIMENTS

This section primarily covers the experimental results and comparison of the following tasks: (i) 4D generation, (ii) 4D reconstruction and (iii) Motion Transfer. Detailed descriptions of the benchmarks (Sec. 6.3), and implementation (Sec. 6.4) are in the Appendix. Additional visual results can be found on our anonymous webpage: `https://magicpose4d.github.io/` and Google Drive: link.

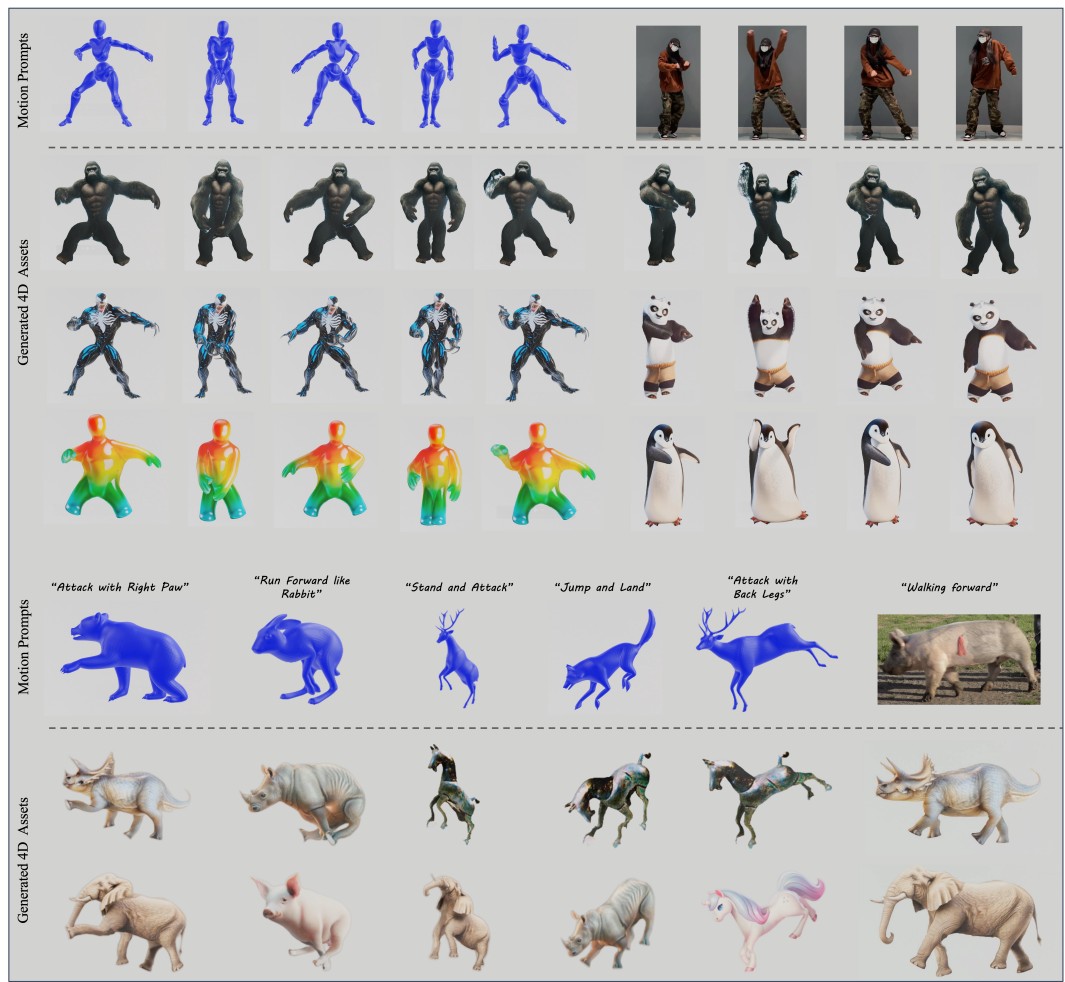

Figure 4: **Appearance and Motion Controlled 4D Generation.**

### 4.1 4D GENERATION

Current 4D generation methods can be categorized into the following types based on how motion is controlled: (1) using text to describe the desired motion; (2) using video to specify the motion, and (3) lacking motion control or only allowing limited path control as shown in Tab.4.

The first category includes methods like SVD Blattmann et al. (2023b)+Stag4D Zeng et al. (2024) (which uses text to control SVD to generate videos with the desired motion and use Stag4D for reconstruction), 4Dfy Bahmani et al. (2024b), Animate124 Zhao et al. (2023), AYG Ling et al. (2023). The second category includes methods such as MagicPose4D and DMT Yatim et al. (2024)+Stag4D

Table 1: User study of MagicPose4D on **4D Generation** compared to 4Dfy, DMT+Stag4D, and SVD+Stag4D. Criteria for judgment: (1) Appearance matches appearance prompts, (2) Motion matches text description, and (3) The overall generation quality. S4D is short for Stag4D. Prompts of each experiment are shown in Tab.5.

| Method | Exp-1 | Exp-2 | Exp-3 | Exp-4 | Exp-5 | Exp-6 | Exp-7 | Exp-8 | Exp-9 | Exp-10 | Exp-11 | Exp-12 | Exp-13 | Exp-14 | Exp-15 | Ave. |
|---|---|---|---|---|---|---|---|---|---|---|---|---|---|---|---|---|
| MP4D | 5.04 | 4.97 | 5.17 | 5.14 | 5.01 | 5.05 | 4.81 | 5.01 | 5.16 | 5.00 | 5.02 | 5.02 | 4.89 | 4.94 | 4.64 | **4.99** |
| 4Dfy | 3.07 | 2.83 | 3.98 | 2.75 | 3.55 | 2.82 | 3.27 | 3.03 | 3.23 | 3.25 | 3.00 | 2.89 | 3.16 | 2.97 | 2.75 | 3.09 |
| DMT+S4D | 2.61 | 2.44 | 1.78 | 2.08 | 2.13 | 2.62 | 2.63 | 2.84 | 2.37 | 2.73 | 2.60 | 2.41 | 2.64 | 2.67 | 2.37 | 2.52 |
| SVD+S4D | 3.25 | 2.86 | 3.40 | 3.29 | 3.08 | 3.26 | 3.14 | 3.31 | 3.35 | 3.31 | 3.22 | 3.02 | 3.03 | 3.31 | 2.94 | 3.19 |

| Method | Exp-16 | Exp-17 | Exp-18 | Exp-19 | Exp-20 | Exp-21 | Exp-22 | Exp-23 | Exp-24 | Exp-25 | Exp-26 | Exp-27 | Exp-28 | Exp-29 | Exp-30 | Ave. |
|---|---|---|---|---|---|---|---|---|---|---|---|---|---|---|---|---|
| MP4D | 5.25 | 5.17 | 4.86 | 4.92 | 5.33 | 5.14 | 4.99 | 5.17 | 5.37 | 4.97 | 4.97 | 5.39 | 5.20 | 4.91 | 5.15 | **5.08** |
| 4Dfy | 3.63 | 3.25 | 3.36 | 3.12 | 3.56 | 2.95 | 2.95 | 3.29 | 2.25 | 2.34 | 2.90 | 2.68 | 3.32 | 2.73 | 2.49 | 3.00 |
| DMT+S4D | 2.65 | 2.40 | 2.34 | 2.42 | 2.11 | 3.01 | 2.35 | 2.47 | 1.86 | 2.22 | 2.49 | 1.97 | 2.60 | 2.19 | 2.32 | 2.40 |
| SVD+S4D | 3.69 | 3.05 | 3.15 | 3.20 | 3.08 | 2.85 | 3.97 | 3.19 | 2.59 | 3.67 | 2.49 | 3.32 | 2.07 | 2.43 | 3.31 | 3.12 |

(where DMT enables video-level pose transfer). The third category comprises methods like DG4D Ren et al. (2023) and TC4D Bahmani et al. (2024a), which provide partial or no motion control.

As shown in Fig. 4, MagicPose4D takes either monocular videos or mesh sequences as motion prompts, enabling precise control over the motion of generated 4D articulated assets. Additional results are presented in the videos provided in the supplementary material. To highlight the advantages of MagicPose4D, we compared it with state-of-the-art (SOAT) methods that allow motion control from each category. However, because current SOAT methods mainly showcase simple, common, imprecise, small-scale, and short-duration motions ($1 - 2$ seconds), they fail to adequately assess a model's ability to control motion. Therefore, we designed tests using more complex, uncommon, precise, large-scale, and long-duration motions (up to 20 seconds), such as "run like a rabbit" (instead of just "run") and "dancing hip hop" (300 frames motion prompts) to better evaluate motion control.

To compare with representative recent works 4Dfy Bahmani et al. (2024b) and Animate124 Zhao et al. (2023), we feed the reference image, which provides the identity, and text prompt, which describes the motion of the generated object (4Dfy only uses text prompts), into their model and optimize the NeRF with SDS-loss. To compare with Stag4D, we first leverage SVD Blattmann et al. (2023b) and DMT Yatim et al. (2024) to generate video according to the input reference text or video and then use Stag4D Zeng et al. (2024) to do reconstruction from the generated video. In our case, we use the reference image to generate the target mesh with an Image-to-3D model and transfer the motion of the reference mesh sequence, which is built by our 4D reconstruction module. We visualize the rendered video from different viewpoints from those SOTA methods and compare them with MagicPose4D in Fig. 5(a) and Fig. 8. More video comparisons are in the supplementary material and on our web page. MagicPose4D provides more temporally consistent and smooth generation with the learning of motion deformation. Since there are no ground truth mesh sequences to evaluate the generation, we provide a comprehensive user study for 4D Generation for a qualitative evaluation in Sec. 6.5 and Tab.1 and 6 to conclude our findings.

## 4.2 4D Cross-Category Motion Transfer

MagicPose4D is able to learn and retarget the motion from dynamic mesh prompts. In this section, we present motion transfer results from MagicPose4D in Fig. 4 and user study in Tab.3. The target identities and reference motion sequences can either be humanoid or animal subjects.

We further compare the motion transfer ability for mesh generation quality with recent methods 3D-CoreNet Song et al. (2021) and X-DualNet Song et al. (2023) in Fig. 5 and Tab.3 (c). Both methods use a deep neural network to learn latent shape codes to retarget the motions and are trained/evaluated on SMPL Loper et al. (2023b)(humanoid) and SMAL Zuffi et al. (2017)(animal). Since they do not include disentangled components or shape deformation modules to represent the structural information of 3D meshes, these methods rely heavily on pre-processed training data with high-quality mesh annotations. Due to these constraints, these baselines cannot generalize well to in-the-wild target identities or reference motions. In contrast, MagicPose4D generates temporally consistent and smooth motions, while strictly preserving the identity and appearance of the target mesh. It is also worth noting that because previous methods focus only on pose transfer without considering the motion trajectory, the generated object always stays in the same position. This can be observed in side-by-side video comparisons on our anonymous webpage: `https://magicpose4d.github.io/`.

Table 2: **2D Keypoint Transfer Accuracy.** This table presents the 2D keypoint transfer accuracy on DAVIS and PlanetZoo datasets. For all methods, we carry out multiple executions and report the mean of the observed accuracy in each case. Note that all experiment results of MP4D are without the skeleton templates and learning underlying structure following Zhang et al. (2024b).

| Method | DAVIS | | | | | PlanetZoo | | | | | | |
|---|---|---|---|---|---|---|---|---|---|---|---|---|
| | camel | cow | dog | bear | dance-twirl | giraffe | tiger | elephant | bear | zebra | deer | Ave. |
| ViSER Yang et al. (2021c) | 71.7 | 73.7 | 65.1 | 72.7 | 78.3 | 51.2 | 68.4 | 68.9 | 60.3 | 55.7 | 57.3 | 65.8 |
| LASR Yang et al. (2021b) | 75.2 | 80.3 | 60.3 | 83.1 | 55.3 | 56.3 | 70.4 | 69.5 | 63.1 | 57.4 | 60.3 | 66.5 |
| LIMR Zhang et al. (2024c) | 77.5 | 79.6 | 65.4 | 84.7 | 79.6 | 59.3 | 64.9 | 68.3 | 64.1 | 57.1 | 55.2 | 68.7 |
| S3O Zhang et al. (2024b) | 79.8 | 83.2 | 67.3 | 86.7 | 84.7 | 63.1 | 72.8 | 71.3 | 66.4 | 61.5 | 62.1 | 72.6 |
| MP4D (Ours) | **80.4** | **84.8** | **69.7** | **87.9** | **86.0** | **64.7** | **75.2** | **72.4** | **68.9** | **61.9** | **63.7** | **74.2** |

## 4.3 4D RECONSTRUCTION

Since the primary goal of our 4D reconstruction module is to better capture motion, we focus on the accuracy of keypoint reconstruction. Therefore, we use keypoint transfer accuracy as the quantitative evaluation metric. We examine the performance of MagicPose4D against the SOTA methods, S3O Zhang et al. (2024b), LIMR Zhang et al. (2024c), LASR Yang et al. (2021a) and ViSER Yang et al. (2021c). As shown in Tab.2, MagicPose4D consistently outperforms them for all animal subjects with a notable margin. When we run the author-provided codes for LASR, and ViSER on DAVIS, and PlanetZoo, we find that their results are highly variable across different runs. The results we report for the mean accuracies we have obtained over multiple runs (5) of all methods. Furthermore, as depicted in Fig.7, we compare the results of MagicPose4D with LASR Yang et al. (2021a) and BANMo Yang et al. (2022a). These methods fall short of achieving ideal reconstruction, due to a lack of structural information about the objects, since they learn from only a single monocular video, which contains sparse views. NeRF-based methods such as BANMo Yang et al. (2022b) and MagicPony Wu et al. (2023) require particularly dense views from multiple cameras or long videos to achieve decent results, while ours achieves good reconstructions from only sparse views.

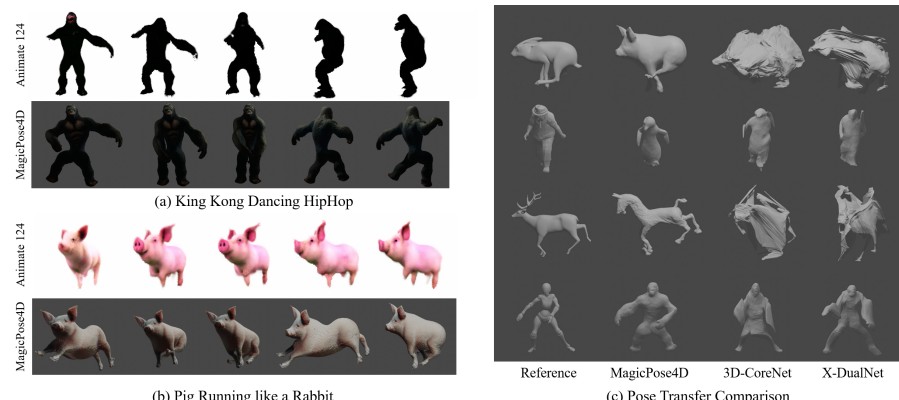

Figure 5: **4D Generation** Comparison of MagicPose4D with Animate124 Zhao et al. (2023), **Motion Transfer** Comparison of MagicPose4D with 3D-CoreNet Song et al. (2021) and X-DualNet Song et al. (2023). Videos are in Sec.6.2.

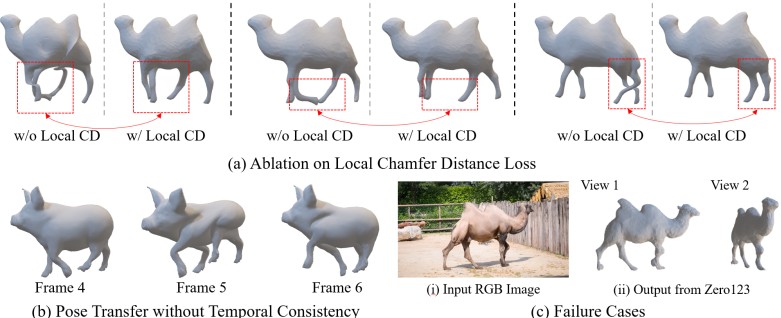

Figure 6: **Ablation Experiments.**

Table 3: User study of MagicPose4D on **Motion Transfer**. We collect the rating results from 50 participants of eleven mesh-sequence comparison experiments. The scale of rating is 0 (low) - 5 (high). Criteria for judgment: **1)** The generated motion should match the reference pose mesh sequence. **2)** The identity of the transferred mesh should match the identity reference. **3)** The generated mesh sequence should be consistent and smooth. More details can be found in Appendix Sec.6.5

| Method | Exp-1 | Exp-2 | Exp-3 | Exp-4 | Exp-5 | Exp-6 | Exp-7 | Exp-8 | Exp-9 | Exp-10 | Exp-11 | Ave. |
|---|---|---|---|---|---|---|---|---|---|---|---|---|
| 3D-CoreNet Song et al. (2021) | 1.85 | 1.73 | 2.10 | 2.31 | 1.90 | 1.73 | 2.29 | 2.73 | 2.65 | 1.88 | 1.79 | 2.09 |
| X-DualNet Song et al. (2023) | 2.88 | 1.94 | 1.53 | 1.88 | 2.22 | 2.18 | 2.14 | 2.86 | 2.71 | 2.22 | 2.31 | 2.26 |
| MP4D(Ours) | **4.92** | **4.47** | **4.69** | **4.39** | **4.45** | **4.45** | **3.90** | **4.12** | **4.02** | **4.10** | **4.20** | **4.34** |

## 4.4 DIAGNOSTIC

**Effectiveness of Global-Local Chamfer Loss.** Relying solely on the geometric information contained in a short monocular video often makes it challenging to achieve satisfactory 3D reconstruction results. This difficulty arises because most videos typically offer sparse viewpoints and the presence of non-rigid object deformations increases the optimization space. Consequently, many methods that rely only on 2D supervision Yang et al. (2021b; 2022b); Wu et al. (2023) struggle to ensure effectiveness on in-the-wild videos. To enhance the generalization capability of the reconstruction module, we utilize existing image-to-3D model methods to obtain geometric priors. By using these models, we generate pseudo-3D meshes for each frame as 3D supervision. We employ the Chamfer distance as the 3D loss objective to align the predicted 3D mesh closely with the pseudo-3D mesh in terms of point distribution. **However**, global Chamfer distance alone can only ensure similar point distributions between the two meshes and does not guarantee the correspondence of points as expected. For instance, as shown in Fig.6(a), without using the local Chamfer loss, the overall point distribution might be similar, but the mesh shape could be entirely incorrect, such as the left and right legs being swapped or vertices originally on the right front leg being moved near the left hind leg. By incorporating the local Chamfer loss, we enforce consistency between each part of the predicted mesh and the pseudo mesh, thus ensuring pose consistency and significantly mitigating the previously mentioned issues.

**Temporal Consistency** is crucial for 4D generation. As shown in Fig.6 (b), performing pose transfer independently for each frame results in noticeable discontinuities when viewing the entire video. To address this, we first define the canonical space using the initial frame, where all frames share the same canonical mesh, skeleton, and skinning weights. Each subsequent frame is derived from the deformation of the first frame using global-local transformations and blending skinning. This approach ensures the consistency of the static model and rigging. Additionally, we introduce a dynamic rigidity regularization term between consecutive frames, which minimizes the deformation of the object between successive frames. This ensures temporal smoothness of the 4D content.

## 5 CONCLUSION & LIMITATIONS

We introduce MagicPose4D, a novel framework for 4D generation providing more accurate and customizable 4D motion transfer. We propose a dual-phase reconstruction process that initially uses accurate 2D and pseudo 3D supervision without skeleton constraints and subsequently refines the model with skeleton constraints to ensure physical plausibility. We incorporate a novel loss function that aligns the overall distribution of mesh vertices with the supervision and maintains part-level alignment without additional annotations. MagicPose4D enables cross-category motion transfer using a kinematic-chain-based skeleton, ensuring smooth transitions between frames through dynamic rigidity and achieving robust generalization without the need for additional training.

The main **limitations** of our method and existing works are: **(i)** Deformation based on blend skinning relies on accurate and robust skeletons and skinning weights predictions, facing a trade-off between generalization and accuracy. Learning-based methods have limited generalization due to restricted training datasets, while non-learning methods suffer from inductive bias, leading to suboptimal results. **(ii)** MagicPose4D can infer poses quickly for pose transfer without training, but 4D reconstruction requires significant training (5 hours on an L40S). **(iii)** Our method struggles with detailed motion control, such as fingers and facial features, due to the challenge of capturing fine-grain details during 4D reconstruction. These issues represent future research directions.

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

Table 4: Support Prompts Comparison. Random represents no control of motion.

| Method | Appearance | | Motion | | | |
|---|---|---|---|---|---|---|
| | Text | Image | Text | Video | Trajectory | Random |
| Animate124 Zhao et al. (2023) | ✓ | ✓ | ✓ | | | |
| 4D-fy Bahmani et al. (2023) | ✓ | | ✓ | | | |
| DreamGaussian4D Ren et al. (2023) | ✓ | ✓ | | | | ✓ |
| AYG Ling et al. (2023) | ✓ | | ✓ | | | |
| Dream-in-4D Zheng et al. (2023) | ✓ | ✓ | ✓ | | | |
| TC4D Bahmani et al. (2024a) | ✓ | ✓ | ✓ | | ✓ | |
| MagicPose4D | ✓ | ✓ | ✓ | ✓ | ✓ | |

# 6 APPENDIX

In this section, we aim to provide additional information not included in the main text due to length constraints. This supplemental content includes:

1. Support prompts comparison with existing 4D generation methods in Tab.4.

2. Key Differences Between the Two Stages in 4D Reconstruction Module in Sec.6.1.

3. Video results (Sec.6.2)

4. Implementation details (Sec.6.4)

5. Information about the dataset used for evaluation (Sec.6.3)

6. In-depth explanation of loss and regularization terms (Sec.6.6)

7. Further experimental results: (i) comparison with existing 4D reconstruction methods in Fig.7, 4D generation method in Fig.8 and Tab.6, (ii) comparison with existing pose transfer methods in Fig.9, (iii) skeleton extraction without skeleton template in Fig.10, and (iv) skeleton and skinning weights results in Fig.11.

8. Details of user study (Sec.6.5)

9. Broader social impacts (Sec.7)

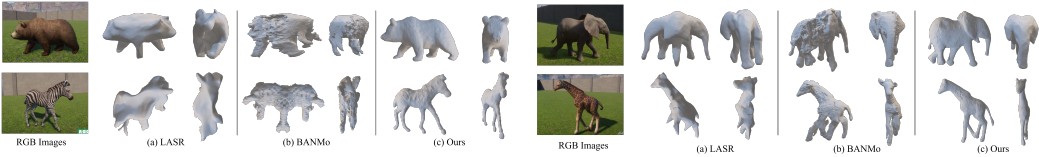

Figure 7: **4D Reconstruction Results.** We show the mesh reconstruction results of (a) LASR, (b) BANMo, and (c) Ours in the PlanetZoo's `bear`, `zebra`, `elephant`, and `giraffe`.

## 6.1 KEY DIFFERENCES BETWEEN THE TWO STAGES IN 4D RECONSTRUCTION

In the first stage of reconstruction, similar to many existing methods [38, 31, 34], each bone can freely perform SE(3) transformations, making it challenging to ensure the physical plausibility of the internal skeleton. In experiments, we observed that while the output mesh was correct, the skin weights and bone transformations were not physically plausible for each variable. Additionally, the obtained bone transformations could not be directly used for motion transfer because the shapes, sizes, and part proportions between the reference object and the target object could differ significantly. Directly applying the same SE(3) transformation to corresponding parts would be unreasonable and would not ensure pose similarity. Therefore, in the second stage of reconstruction, we use kinematic chain skeletons and apply heat diffusion to obtain skinning weights, which narrow the deformation space of the skeleton, thereby ensuring the plausibility of the internal structure. This method uses forward kinematics to transfer pre-frame bone angles and bone scales, ensuring accurate motion (pose) transfer.

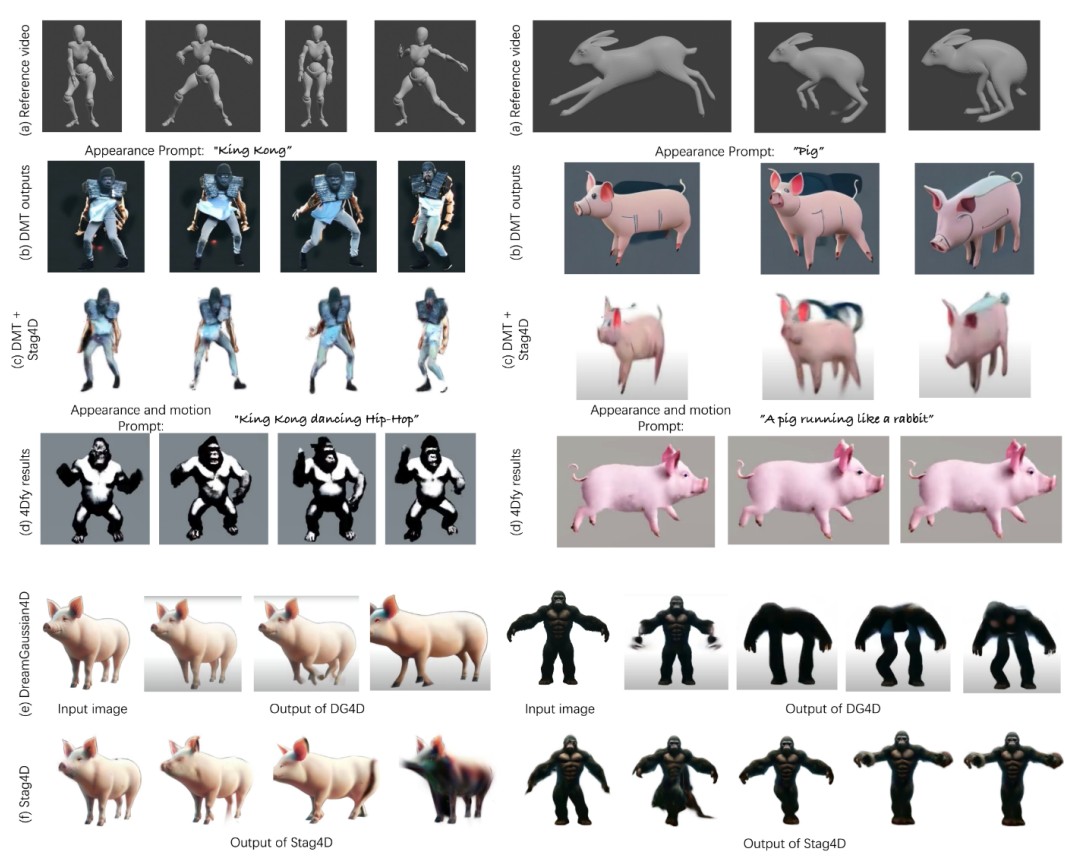

Figure 8: **4D Generation** Comparison of MagicPose4D with 4Dfy, Stag4D, DG4D, DMT+Stag4D, and Animate124.

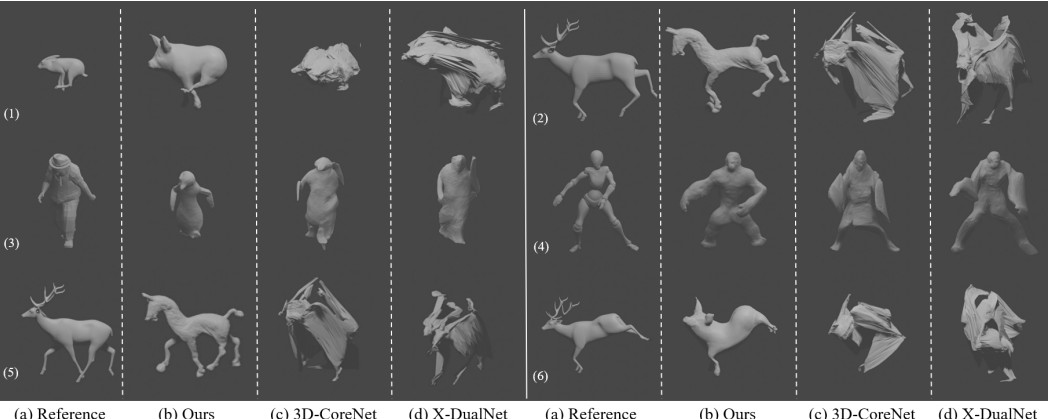

Figure 9: **Motion Transfer Comparison with 3D-CoreNet Song et al. (2021) and X-DualNet Song et al. (2023)**

## 6.2 VIDEO RESULTS

We include an extensive set of 4D generation results in video format in Google Drive: link. There, we demonstrate 4D generation results across different species with diverse motions and compare the

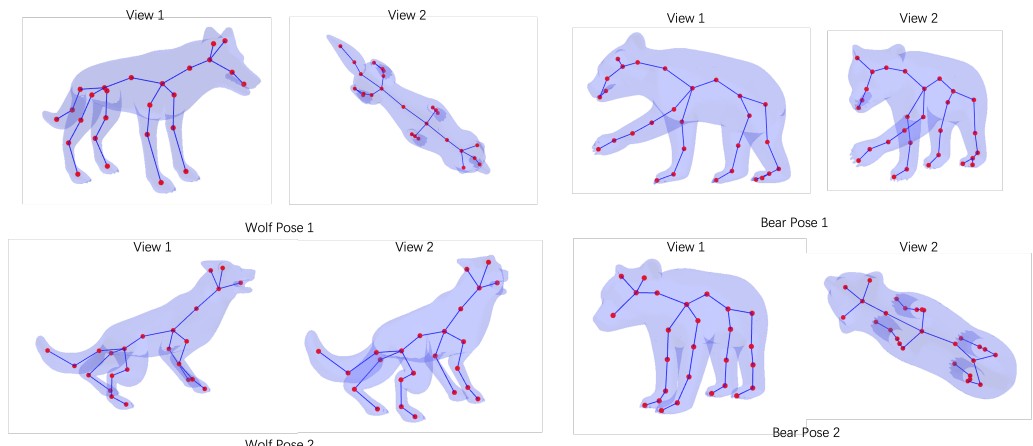

Figure 10: **Generated skeleton result without skeleton template.**

performance of pose transfer with 3D-CoreNet Song et al. (2021), X-DualNet Song et al. (2023), 4Dfy, SVD+Stag4D, DMT+Stag4D and Animate124 Zhao et al. (2023). Since the video files exceed 1000 MB, we are unable to upload them directly as a zip file. Therefore, we have used an anonymous Google Drive link.

Table 5: Prompts for User Study of 4D Generation.

| # | Prompts |
|---|---|
| 1 | A pig running like a rabbit |
| 2 | A pig attacking with its head |
| 3 | King Kong dancing hip hop |
| 4 | A pig dies and fall down |
| 5 | A pig jumping high and landing |
| 6 | A pig walking back |
| 7 | A pig jumptorting like a deer |
| 8 | Kung Fu Panda dancing hip hop |
| 9 | A penguin dancing hip hop |
| 10 | King Kong drunk walking |
| 11 | Kung Fu panda drunk walking |
| 12 | A penguin drunk walking |
| 13 | A pig attacking using its hind legs |
| 14 | A pig torting like a deer |
| 15 | A pig run like a tiger |
| 16 | King Kong hands up |
| 17 | Kung Fu panda hands up |
| 18 | A pig head down and drinking water |
| 19 | A pig stands and attacks with its left hand |
| 20 | A pig jumps on a box like a dog |
| 21 | A jelly man dancing hip hop |
| 22 | A Venom dancing hip hop |
| 23 | A unicorn attacking using its hind legs |
| 24 | A unicorn torting like a deer |
| 25 | A elephant run like a rabbit |
| 26 | A jelly man drunk walking |
| 27 | A Venom drunk walking |
| 28 | A rhino runs like a rabbit |
| 29 | A elephant stands and attack |
| 30 | A triceratops attack with its right leg |

### 6.3 DATASET

#### 6.3.1 ANIMAL

**Davis-Camel** provides a real animal video in BADJA Biggs et al. (2018) with 2D keypoints and mask annotations, derived from the DAVIS video segmentation dataset Perazzi et al. (2016) and online stock footage. We extract reference motion from the reconstructed mesh sequence and transfer it to other identities.

**PlanetZoo** includes RGB synthetic videos of different animals with around 100 frames each. PlanetZoo covers a 180-degree visual field captured by a moving camera to allow better evaluation of 3D reconstruction when imaging parameters must also be dynamically estimated due to the moving camera, and over a large visual field. In addition, following BADJA Biggs et al. (2018), we also provide 2D key point annotations.

**DeformingThings4D** is a synthetic dataset containing 1,972 animation sequences containing 31 categories of both humanoids and animals. Each sequence consists of 40 to 120 frames of motion animation. In this dataset, the first frame is the canonical frame, and its triangle mesh is given. From the 2nd to the last frame, the 3D offsets of the mesh vertices are provided, and we export the triangle meshes for all these frames. We use the motions of these animal mesh sequences as pose references and transfer the pose to different identities.

#### 6.3.2 HUMAN

**EverybodyDanceNow** consists of **full-body** videos of five human subjects. We use these monocular videos to generate human motions and transfer them to other identities.

**DeformingThings4D** also contains humanoid examples of dynamic mesh, as mentioned before. We use the motions of these sequences as pose references and transfer the pose to different identities.

#### 6.3.3 SELF-COLLECTED IN-THE-WILD DATA

These in-the-wild data are collected from online resources, like the video of HipHop dancing from Youtube.

### 6.4 IMPLEMENTATION DETAILS

For Davis-Camel, PlanetZoo, EverybodyDanceNow, and those self-collected data without ground truth mesh, we first train the Dual-Phase 4D Reconstruction Module on 2 NVIDIA L40S GPUs with batch size 16 for 10 epochs with a learning rate of 0.0001. MagicPose4D uses Zero123 Liu et al. (2023) and SF3D Boss et al. (2024) as the image-to-3D model for 4D reconstruction. We then transfer the per-frame motion from Phase 2 of 4D Reconstruction to the target object with the Cross-Category Motion Transfer Module. The motion transfer is not learning-based and does not require any trainable parameters, which makes MagicPose4D generalize well to unseen identities and reference motions. For all other data with ground truth mesh available, we directly train the second phase of the 4D Reconstruction Module and then transfer the motions.

### 6.5 USER STUDY

We provide a user study for comparison between MagicPose4D and previous works Song et al. (2021; 2023) on motion transfer. We asked 50 lay participants from **Prolific**, an online platform for user studies, to rate the quality of **eleven** in-the-wild retargeted mesh sequences from 3D-CoreNet Song et al. (2021), X-DualNet Song et al. (2023) and MagicPose4D on a scale of 0(low) to 5(high). The participants are paid with an hourly rate of 16 USD. In each mesh sequence comparison, we collect target identity mesh, reference motion mesh sequence, and retargeting results. We visualize them side-by-side. The retargeting results from different methods are anonymized as A, B, C, and the order is randomized. We provide video visualization of these comparisons (**For each video, from left to right: Reference Motion; MagicPose4D; 3D-CoreNet; X-DualNet**) in the Google Drive. Criteria for judgment: **1)** The generated motion should match the reference pose mesh sequence. **2)** The identity of the transferred mesh should match the identity reference. **3)** The generated mesh

Table 6: User study of MagicPose4D on 4D Generation compared to Animate124 Zhao et al. (2023). The settings are the same as the study on Motion Transfer.

| Method | Exp-1 | Exp-2 | Exp-3 | Exp-4 | Exp-5 | Exp-6 | Ave. |
|---|---|---|---|---|---|---|---|
| Animate124 Zhao et al. (2023) | 2.60 | 1.98 | 1.90 | 1.92 | 2.28 | 2.24 | 2.15 |
| MagicPose4D | **4.53** | **4.37** | **3.91** | **4.00** | **4.60** | **4.62** | **4.34** |

sequence should be consistent and smooth. From the results presented in Tab. 3, we conclude that MagicPose4D provides the most satisfying generation of mesh sequences.

Similarly, we provide a user study for comparison with 4Dfy, DMT+Stag4D, and SVD+Stag4D. In each comparison experiment, we collect reference images, text descriptions of motion, mesh reference motion sequences, and generated videos from both methods. For 4Dfy we follow the official code and instructions, for the other two, we first use SVD to generate a video via text prompt or use DMT to do video-level motion transfer and then use Stag4D to do reconstruction from the generated video. For MagicPose4D, we use the reference image to generate the target mesh with an Image-to-3D model and transfer the motion of the reference mesh sequence. We compared the rendered videos from Animate124 to videos of generated mesh from MagicPose4D in different viewpoints. Criteria for judgment: **1)** The generation's identity should match the reference image. **2)** The generated mesh sequence should be consistent and smooth. From the results presented in Tab. 1, we conclude that MagicPose4D provides the most satisfying visualizations.

Furthermore, we conduct a user study for comparison with STAG4D Zeng et al. (2024), 4Dfy Bahmani et al. (2024b), on 4D generation of articulated objects. We asked 50 participants from Prolific to rate the quality of nine sequences (3 objects from 3 views) on a scale of 0(low) to 5(high). In each sequence comparison, we collect reference images, text descriptions of motion sequences, and video results. We visualize them side-by-side. The video results from different methods are anonymized as A, B, and C, and the order is randomized. The Criteria for Judgment: 1) The overall generation quality - completeness; smoothness; consistency; etc. 2) The motion of the object in the video should match the text description. 3) The appearance of the object in the video should match the reference image. The settings for the participants are the same. We provide video visualization of these comparisons in the Google Drive.

## 6.6 2D LOSSES AND REGULARIZATION

The 3D loss is described in the main article. Here we introduce the 2D losses and regularization terms. The 2D losses are similar to those in existing differentiable rendering pipelines Yang et al. (2021b; 2022b); Zhang et al. (2024c). We define $\mathbf{S^t}, \mathbf{I^t}, \mathbf{F}^{2D,t}$ as the silhouette, input image, and optical flow of the input image, and their corresponding rendered counterparts as $\widetilde{\mathbf{S}}^{\mathbf{t}}, \widetilde{\mathbf{I}}^{\mathbf{t}}, \widetilde{\mathbf{F}}^{2D,t}$. The following losses ensure the fitting between rendered and original:

$$\mathcal{L}_{\text{silhouette}} = \sum_{\mathbf{x}^t} \left\| \mathbf{s}(\mathbf{x}^t) - \hat{\mathbf{s}}(\mathbf{x}^t) \right\|_2, \tag{4}$$

$$\mathcal{L}_{\text{optical flow}} = \sigma \left\| (\widetilde{\mathbf{F}}^{\mathbf{2D,t}}) - (\mathbf{F}^{2D,t}) \right\|_2^2, \tag{5}$$

$$\mathcal{L}_{\text{texture}} = \left\| \widetilde{\mathbf{I}}^{\mathbf{t}} - \mathbf{I}^{\mathbf{t}} \right\|_1, \tag{6}$$

$$\mathcal{L}_{\text{perceptual}} = \text{pdist}(\widetilde{\mathbf{I}}^{\mathbf{t}}, \mathbf{I}^t), \tag{7}$$

where pdist(,) is the perceptual distance Zhang et al. (2018). Also, we leverage three **regularization terms:** (1) Dynamic Rigidity term, which is introduced in paper.Zhang et al. (2024c). (2) Symmetry term, which encourages the canonical mesh to be symmetric.

$$\mathcal{L}_{\text{symm}} = \sum_i \min_j \| v_j - \phi \left( v_i{}^{symm} \right) \|^2, \tag{8}$$

where, $v_j$ is the vertex j in the one side of the symmetry plane, and $v_i{}^{symm}$ is the vertex from the other side. $\phi$ is the reflection operation w.r.t to the symmetry plane. (3) Laplacian smoothing: We apply laplacian smoothing to generate smooth mesh surfaces.

$$\mathcal{L}_{\text{shape}} = \left\| \mathbf{X}_i^0 - \frac{1}{|N_i|} \sum_{j \in N_i} \mathbf{X}_j^0 \right\|^2 , \tag{9}$$

where, $\mathbf{X}_i^0$ is coordinates of vertex $i$ in canonical space. And $N$ is the number of vertices

## 7 BROADER SOCIAL IMPACTS

The proposed MagicPose4D for motion transfer offers extensive applications, enhancing communication in digital environments by enabling more effective self-expression through avatars or digital characters. Additionally, MagicPose4D has the potential to revolutionize the entertainment and media production industries by facilitating the creation of more lifelike and expressive characters in movies, video games, and animations.

However, this technology also presents potential negative social impacts. Privacy concerns arise from the unauthorized creation of realistic animations of individuals, and the technology could facilitate the spread of misinformation through deepfakes. Risks include job displacement in fields like acting and modeling, psychological effects from the blurring of reality and virtual experiences, and the exploitation of the technology for unethical purposes. Furthermore, cultural insensitivity, security threats, and the misuse of realistic animal animations could have broader societal implications. Addressing these issues requires robust ethical guidelines, legal frameworks, and technological safeguards to ensure responsible use and mitigate harm.

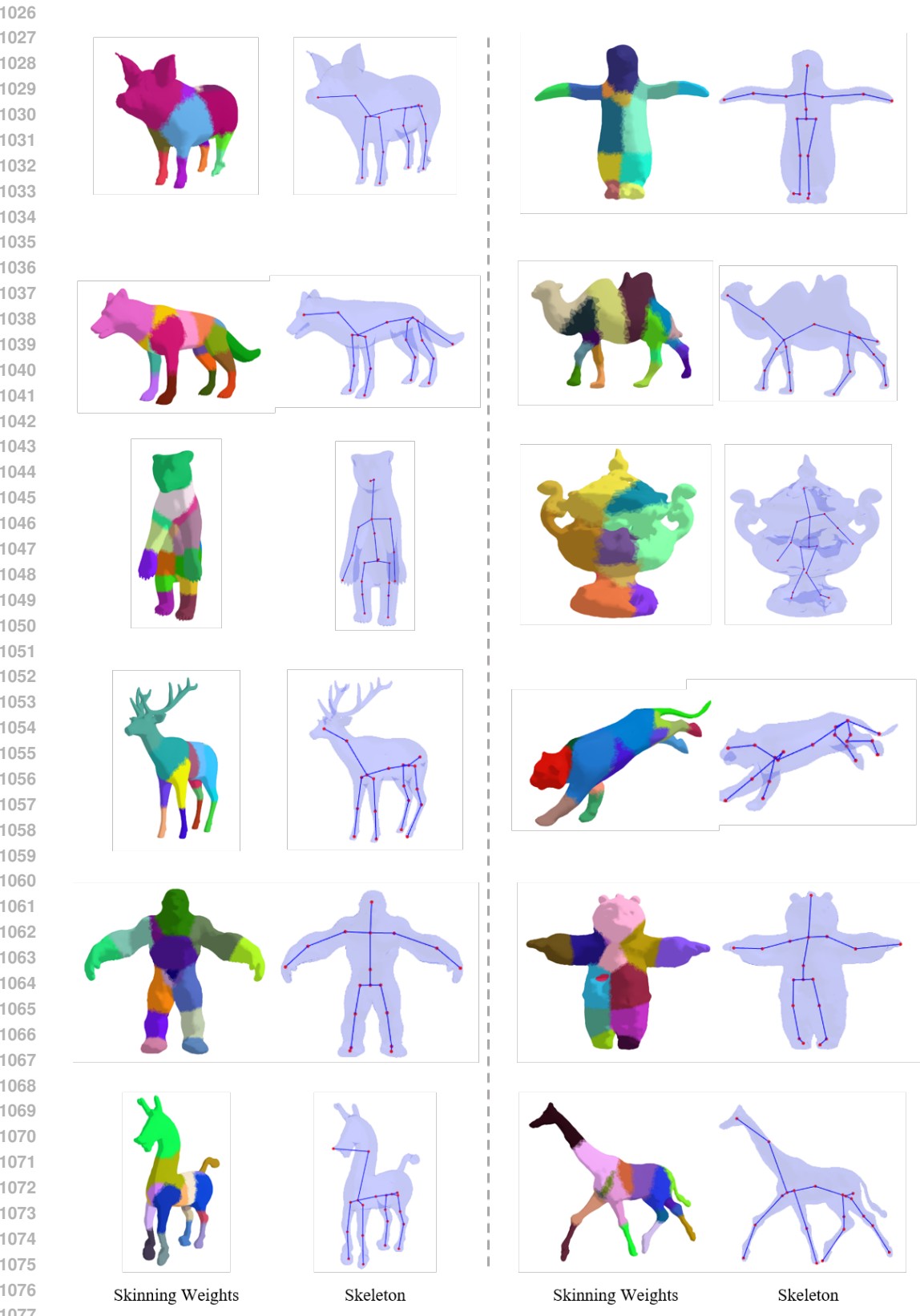

Skinning Weights      Skeleton        Skinning Weights      Skeleton

Figure 11: **Skinning Weights** and **Skeleton** Results from Our Method.

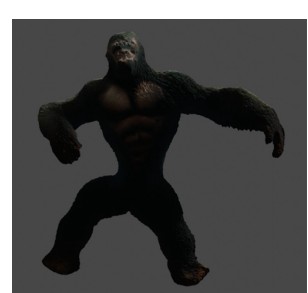 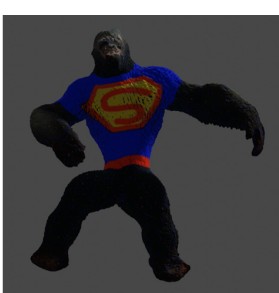 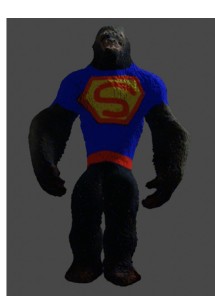 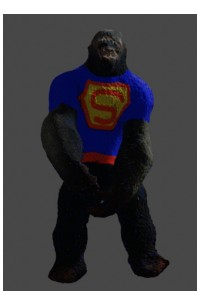

Figure 12: **Post Texture Editing.**

