# OpenReview forum: "MagicPose4D: Crafting Articulated Models with Appearance and Motion Control"
_ICLR.cc/2025/Conference — ICLR 2025 Conference Withdrawn Submission_

### Official Review · Reviewer_MFZd · 2024-10-22

**Soundness:** 3
**Presentation:** 2
**Contribution:** 3
**Rating:** 5
**Confidence:** 3

**Summary:**

This paper proposes MagicPose4D that accepts monocular videos or mesh sequences as motion prompts for refined control over both appearance and motion in 4D generation problem. MagicPose4D comprises a dual-phase 4D reconstruction module that first use 2D and pseudo 3D supervision to capture the model's shape appearance, and subsequently refines the model with kinematic chain-based skeleton constraints to ensure physical plausibility. To ensure smooth transitions between frames, MagicPose4D uses a kinematic-chain-based skeleton to achieve cross-category motion transfer, ensuring dynamic rigidity and achieving robust generalization without the need for additional training.

**Strengths:**

1. The overall pipeline is very effective, demonstrated by both quantitative and qualitative results that MagicPose4D significantly improves the accuracy and consistency of 4D content generation, outperforming existing methods in various benchmarks.
2. The motivation of accepting monocular videos or mesh sequences as motion prompts is promising, as it can enable precise and customizable motion control.
2. The usage of global-local chamfer loss is to ensure that the predicted mesh closely resembles the expected mesh is novel, since it help align the overall distribution of mesh vertices with the supervision and maintains part-level alignment without additional annotations.

**Weaknesses:**

1. The figures in the paper are not informative enough, making readers a little bit confused when first saw them. I would expect more concise description of each component in these figures in their captions.
2. The author mentions that directly applying image-to-3D model to each frame of the video cannot handle issues like self-occlusion and temporal continuity and smoothness, but I did not see any analysis on how Magic4D impose temporal consistency. Can the supervision and losses applied alleviate this issue?
3. Pseudo-3D ground truth seems like a crutial part in the supervision, I'm wondering what if the image-to-3D model fails and the predicted mesh is unsatisfactory, will it be harmful to the learning process?
4. In Fig.3(a), the skeleton template is from human, but the reference subject is a camel, I'm wondering how to deform a human skeleton and embed it in a completely non-relevant species. The technical details of model articulation part is too few in the paper, I would prefer the author to spend more space in this section.
5. In section 4.3, the performance regarding 2D Keypoint Transfer Accuracy not does not outperform S3O notably according to Table 2, can the author provide further analysis of the advantages of MP4D compared with S3O?
6. In the demo video provided in the supplementary material, the results are not as good as I expected. The characters are a little bit distorted and I'm wondering why the root trajectories are not stable.
6.

**Questions:**

Please refer to previous section

---

### Official Review · Reviewer_TjpL · 2024-11-03

**Soundness:** 2
**Presentation:** 3
**Contribution:** 2
**Rating:** 3
**Confidence:** 5

**Summary:**

The paper introduces MagicPose4D, a framework for 4D content generation that enables refined control over both the appearance and motion of articulated models. It addresses limitations in existing methods by accepting monocular videos or mesh sequences as motion prompts, allowing for precise motion control.

The framework consists of two key modules: the Dual-Phase 4D Reconstruction Module, which captures model shape and motion with a two-phase approach using 2D and pseudo-3D supervision, and the Cross-Category Motion Transfer Module, which facilitates motion transfer across different categories without additional training.

Besides, MagicPose4D introduces a Global-Local Chamfer loss for better mesh vertex alignment and demonstrates improvements in accuracy and consistency over existing methods in 4D content generation.

**Strengths:**

The strengths of MagicPose4D lie in its approach to 4D content generation, which provides enhanced control and precision over the appearance and motion of articulated models. The framework's Dual-Phase 4D Reconstruction Module captures the shape and motion of models using a combination of 2D and pseudo-3D supervision, while the Cross-Category Motion Transfer Module allows for the transfer of motion across different categories without the need for additional training.

Additionally, the introduction of the Global-Local Chamfer loss function improves the alignment of predicted mesh vertices with the supervisory 3D model, maintaining both overall and part-level accuracy.

**Weaknesses:**

Although acknowledged in the paper, the primary weaknesses of MagicPose4D involve the reliance on accurate and robust skeleton and skinning weight predictions for deformation, which presents a trade-off between generalization and accuracy. The method's limited generalization is due to the constraints of training datasets, and non-learning methods may suffer from inductive bias, leading to suboptimal results.

Additionally, while MagicPose4D can quickly infer poses for pose transfer without training, the 4D reconstruction process is resource-intensive, requiring large training time. The framework also struggles with detailed motion control, such as for fingers and facial features, due to the challenges in capturing fine-grain details during 4D reconstruction.

Overall, I think that this pipeline does not offer substantial novelties to the field of 4D generation.

**Questions:**

My first question is about the capabilities of MagicPose4D, particularly its performance on articulated models such as non-human and quadruped subjects. Besides, could you elaborate on how MagicPose4D can be extended or adapted to handle dynamic scenes involving multiple non-human and quadruped subjects?

Specifically, I am interested in understanding how the framework would maintain the physical plausibility and temporal consistency of motion across a broader range of articulated objects within a scene.

---

### Official Review · Reviewer_xjRE · 2024-11-03

**Soundness:** 3
**Presentation:** 3
**Contribution:** 2
**Rating:** 6
**Confidence:** 4

**Summary:**

This paper introduces MagicPose4D, a framework aimed at enhancing 4D content generation. MagicPose4D enables precise control over both appearance and motion by accepting monocular videos or mesh sequences as motion prompts. Key contributions include Dual-Phase 4D Reconstruction Module and Cross-category Motion Transfer Module. The proposed model designs a two-phase approach to capture shape and motion, a Global-Local Chamfer loss is also introduced to align predicted mesh vertices effectively. The proposed model uses a kinematic-chain-based skeleton to transfer motion across categories. The experimental results demonstrate that MagicPose4D improves the accuracy and consistency of 4D content generation, surpassing existing methods across selected benchmarks.

**Strengths:**

1. The overall framework for control over motion in 4D generation is interesting as 4D motion generation is one of the main challenge for 4D generation.
2. Learning the canonical appearance and rigging representation from the motion prompts (e.g. monocular videos) provides one way to model the reference sequence.
3. Extendable Bones can enhances the flexibility and realism of rigid hinge connections of skeletal model.
4. The selected visualizations show the efficiency of the proposed framework.

**Weaknesses:**

1. The proposed method rely on the text-to-3D and image-to-3D pretrained models, the generation quality of prior 3D model will seriously affect the 4D generation of the proposed method. The generation of 3D model is not fully optimized in the training of the framework, only some motions are optimized.
2. It is also challenging to extract skeleton motion references from a given monocular video. If the directly obtained skeleton motions sequences are not good enough, the 4D generation of the proposed method will be seriously affected.
3. Although the paper proposes an extendable bones strategy, the approach is still limited to skeleton-based motion control. The authors should show that their approach is skeleton-based 4D action generation, rather than a freestyle approach that does not rely on parameters-based motion priors.
4. The supervisions is composed of many losses, e.g., silhouette loss, optical flow loss, texture loss, perceptual loss, smooth, motion, and symmetric regularizations, and Global-Local Chamfer (GLC) Loss, etc., which may make the network very difficult to train, and the weights and effects of individual losses are not accurately analyzed.

**Questions:**

1. The contribution of the cross-category motion transfer part can be further illustrated, so far it is just a combination of several methods, i.e., Baran & Popovi´c (2007), Bærentzen & Rotenberg (2021), Zhang et al. (2024c).
2. The comparisons in Fig.5 (c) is not fair, the advantage comes from the quality of the 3D reconstruction, not from the motion control. But 3D reconstruction is not the main innovation of the paper.

---

### Official Review · Reviewer_D5wT · 2024-11-04

**Soundness:** 4
**Presentation:** 4
**Contribution:** 3
**Rating:** 5
**Confidence:** 4

**Summary:**

This paper proposes a novel framework for 4D generation, which can accpet video as input and generate consistent motion for 3D mesh. To be more specific, the dual-phase 4D reconstruction module can reconstruct reference motion from video motion prompts. The cross-category motion transfer module can retarget the reference motion to the target object.

**Strengths:**

The overall framework is well-designed, capable of effectively handling various forms of input. Compared to previous work, it maintains better temporal consistency. Both the qualitative visualizations and quantitative experimental comparisons show significant improvements over previous work.

**Weaknesses:**

As a complex framework that integrates various previous works, it would be beneficial to focus more on highlighting the unique contributions of this study. Overall, the motion quality is below expectations, with visualized results displaying noticeable jitter. It falls short of the smoothness claimed in the paper, and the translation appears inaccurate.

The paper spends considerable time describing how to derive reference motion from video prompts and generate corresponding results. However, it only provides a single dance example, which I find unconvincing. I would like to see more visual results or demonstrations of intermediate processes, such as what the reference looks like from the video, how it performs on videos of quadrupeds, or the results for meshes with significant geometric differences from the template.

In addition, the efficiency of the 4D reconstruction part is too low, making it nearly unacceptable for generative tasks. Therefore, the claimed effectiveness and robustness of using video prompts are questionable and require further clarification.

**Questions:**

As a complex system composed of multiple stages, I think it would be beneficial to present some intermediate results. I'm somewhat concerned about how well the reconstruction from video, claimed as a significant contribution in the paper, actually performs. Perhaps the authors could provide more information on success rates or failure cases.

---

### Note · Authors · 2024-11-16

I have read and agree with the venue's withdrawal policy on behalf of myself and my co-authors.